# STABLE BATCHED BANDIT:
# OPTIMAL REGRET WITH FREE INFERENCE

## ABSTRACT

In this paper, we discuss statistical inference when using a sequential strategy to collect data. While inferential tasks become challenging with sequentially collected data, we argue that this problem can be alleviated when the sequential algorithm satisfies certain stability properties; we call such algorithms stable bandit algorithms. Focusing on batched bandit problems, we first demonstrate that popular algorithms including the greedy-UCB algorithm and $\epsilon$-greedy ETC algorithms are not stable, complicating downstream inferential tasks. Our main result shows that a form of elimination algorithm is stable in the batched bandit setup, and we characterize the asymptotic distribution of the sample means. This result allows us to construct asymptotically exact confidence intervals for arm-means which are sharper than existing concentration-based bounds. As a byproduct of our main results, we propose an Explore and Commit (ETC) strategy, which is stable — thus allowing easy statistical inference— and also attains optimal regret up to a factor of 4.

Our work connects two historically conflicting paradigms in sequential learning environments: regret minimization and statistical inference. Ultimately, we demonstrate that it is possible to minimize regret without sacrificing the ease of performing statistical inference, bridging the gap between these two important aspects of sequential decision-making.

## 1 INTRODUCTION

Reinforcement learning (RL) has emerged as a pivotal paradigm in artificial intelligence, driving significant advancements across diverse domains. Its impact spans from theoretical computer science to practical applications in robotics, control systems, and beyond. At the core of RL lies the fundamental challenge of balancing exploration and exploitation - a dilemma that encapsulates the agent's need to gather new information about its environment while simultaneously leveraging existing knowledge to maximize rewards. This balance is crucial for developing effective decision-making strategies through environmental interaction, positioning RL as a cornerstone technology in the evolution of autonomous systems.

In many real-world applications of reinforcement learning, data is collected sequentially and often in batches, reflecting practical constraints and operational realities. This batched approach to data collection is particularly prevalent in domains such as online education Kizilcec et al. (2020), mobile health interventions Liao et al. (2020); Klasnja et al. (2019); Yom-Tov et al. (2017), and digital marketing Li et al. (2010), where multiple users interact with systems simultaneously. While traditional RL algorithms excel at optimizing performance within a specific problem instance, there is a growing need for methods that can extract generalizable insights from the collected data. Statistical inference on sequentially collected data becomes crucial when the goal extends beyond mere performance optimization to include scientific discovery and informed decision-making for future implementations. Consider a mobile app designed to improve dental hygiene habits Trella et al. (2024); Nahum-Shani et al. (2024). The app uses RL to personalize reminders and brushing technique tips. Beyond maximizing daily app engagement, researchers and dentists would be interested in understanding which interventions most effectively promote long-term oral health improvements. They might want to determine if gamified brushing sessions are more impactful than educational content, or if the frequency of reminders significantly affects adherence to recommended brushing

duration. This knowledge could guide the development of future dental health interventions, allow for refinement of less effective strategies, and contribute to our understanding of habit formation in oral care.

In this paper, we focus on the problem of statistical inference in bandits problems with data collected in batches; colloquially known as batched bandit problems. While bandit strategies focus on minimizing regret, the sequential (non-iid) nature of bandit algorithms make the down-steam statistical inference much more challenging. For instance, sample means maybe biased for bandit data Nie et al. (2018), and the sample means may not be asymptotically normal Zhang et al. (2020); Ying et al. (2024). In the following section, we provide a brief survey of batched bandit algorithms, with a special focus on explore and commit (ETC) strategies, and on statistical inference with the data collected from a sequential procedure, akin to a bandit algorithm.

## 1.1 RELATED WORK

### 1.1.1 BATCHED BANDITS AND EXPLORE-THEN-COMMIT ALGORITHMS

The study of batched bandits has gained significant attention in recent years, with a focus on algorithms that balance exploration and exploitation in a limited number of interaction rounds. Explore-Then-Commit (ETC) algorithms represent a special case of batched bandits where the learning process is divided into two distinct phases: an exploration phase followed by a commitment phase. See the work of Robbins (1952); Anscombe (1963). Perchet et al. (2016) proposed a general strategy for constructing batched bandit algorithms, including ETC-type approaches. Their work addressed the crucial aspect of batch size selection, which may vary across batches to obtain minimax regret bounds. Building on this foundation, Gao et al. (2019) investigated whether adaptively chosen batch sizes could further reduce regret in batched settings. Exploring different aspects of batched bandits, Jin et al. (2021) examined a scenario with a random horizon, ensuring asymptotically optimal regret for exponential families as reward distributions. This work highlighted the flexibility of batched approaches in handling uncertain time horizons. The algorithm that we study in this paper is motivated from the work of Auer & Ortner (2010), where the authors discussed an elimination-based algorithm for batched bandits.

### 1.1.2 STATISTICAL INFERENCE WITH BANDIT DATA

The challenge of performing valid statistical inference with sequentially collected data, particularly in batched bandit settings, has become an important area of research. Zhang et al. (2020) demonstrated that the average reward obtained from batched bandit algorithms is not necessarily asymptotically normal, and proposed a batched OLS estimator for inference in non-stationary settings. To address these challenges, researchers have developed two main approaches: non-asymptotic methods based on concentration bounds for self-normalized martingales Abbasi-Yadkori et al. (2011), and asymptotic methods exploiting the martingale nature of the data and debiasing techniques. See the works in Hall & Heyde (2014); Zhang & Zhang (2014); Khamaru et al. (2021); Ying et al. (2024); Lin et al. (2023); Bibaut et al. (2021); Hadad et al. (2021); Zhang et al. (2021); Abbasi-Yadkori et al. (2011) and references therein.

## 1.2 CONTRIBUTIONS

Our approach to inference in the bandit problem is significantly different from existing approaches. As we already pointed out in the previous related work section, most of the inference methods are *post-processing* methods; meaning they utilize very little information of the bandit algorithm itself, and rely on the Martingale structure present in the sequentially collected data. While this approach is more flexible, the worst-case guarantees for such methods can be pessimistic; see the paper Khamaru et al. (2021); Lattimore (2023) for worst-case lower bounds.

In contrast, we discuss classes of algorithms, which we call *stable bandit algorithms*, where no such post-processing is needed, and classical statistical methods — which are used for iid data — can be used. At a very high level,

*We can treat bandit data as iid data (asymptotically) when the bandit algorithm is stable.*

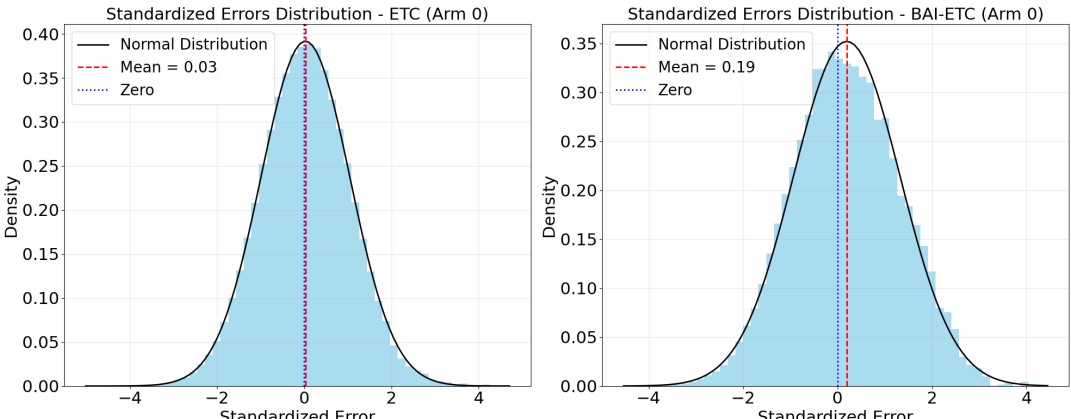

Figure 1: Stable ETC arm 1                    Figure 2: BAI ETC arm 1

Figure 3: Comparison of error distributions for stable-ETC Algorithm 2 and BAI ETC algorithm for a two-armed-bandit with Gaussian rewards and $\mu_1 = \mu_2 = 1$. We see that the asymptotic distribution of the arm-means are close to Gaussian when stable-ETC — *a stable algorithm*— is used. But, the distributions of arm means are not Gaussian when BAI-ETC algorithm — which provides optimal regret — is used; the mean of standardized noise are significantly positive, close to 0.20. We also show (in Corollary 2) that the regret of stable-ETC is no more than 4-times the the optimal-regret. The simulation results are average of 5000 repetitions and the horizon is set to $T = 1000$. See Appendix B for a detailed simulation.

The notion of stable bandit algorithm is motivated from the seminal work of Lai & Wei (1982). To the best of our knowledge, this work is the first to show stable bandit algorithms in batched settings.

Our main contributions are as follows:

- First, we introduce a class of bandit algorithms for multi-armed bandits, which we call *stable bandit algorithms*, and argue that the sample means for each arm are asymptotically normal, when the bandit data is collected using a stable bandit algorithm.

- In Section 3.2 we focus on 2-batch algorithms. We demonstrate that the vanilla $\epsilon$-greedy explore-then-commit (ETC) algorithm is not stable, and we propose a modification of the explore-then-commit algorithm which is stable. An interesting result in this section is a stable ETC algorithm whose regret is optimal up to a factor 4.

- In Section 3.3 we focus on $B$-batch algorithms. In Algorithm 3, we propose a $B$-batch algorithm, in Theorem 2 we discuss the stability property of this algorithm, and characterize the asymptotic distribution of the sample-means.

## 2 PROBLEM SET UP

In this paper we focus on multiarmed bandit algorithm where the data is collected in multiple batches. For sake of exposition, we discuss the two-armed case in full details, though many of our results extend to the K-armed setting. At each round $1 \leq t \leq T$, we select an arm $A_t \in \{1, 2\}$ and receive a reward $Y_t \in \mathbb{R}$ from the distribution $\mathcal{P}_{A_t}$. We assume

- Let $\mu_a$ and $\sigma_a^2$, respectively, denote the mean and variance of the distribution $\mathcal{P}_a$. We assume that $\mathcal{P}_a$ is a sub-Gaussian random variable with sub-Gaussian parameter $\lambda_a$. The parameters $(\mu_a, \sigma_a^2, \lambda_a)$ are unknown and without loss of genrality we assume that $\lambda_a \leq 1$ for $a = 1, 2$.

The focus of this paper is to understand bandit algorithms where the data is collected in batches. We consider two types of batched bandit algorithms:

1. **Two batch algorithm:** In Section 3.2 we focus on a two batch algorithm where the number of arms within each batch goes to $\infty$. See Algorithms 1 and 2 for more details. The algorithm discussed in this section are motivated from Explore Then Commit (ETC) strategies Robbins (1952); Anscombe (1963), and draws inspiration from the ETC type algorithm discussed in Auer & Ortner (2010).

2. $B$-**batch algorithm:** In Section 3, we focus on algorithms where the data is collected in $B$ batches. The number of rounds in each batch, which we denote by $2m$ remains fixed, and we let the number of batches $B$ to $\infty$. We detail our $B$-batch procedure in Algorithm 3

**Goal:** The goal in both cases is to understand the asymptotic properties of the samples means for both arms defined as

$$\bar{\mu}_{a,T} = \frac{1}{n_{a,T}} \cdot \sum_{t=1}^{T} Y_t \cdot \mathbf{1}_{A_t=a} \quad \text{where} \quad n_{a,T} = \sum_{t=1}^{T} \mathbf{1}_{A_t=a}.$$

We are interested to understand the asymptotic behavior of the sample means $(\bar{\mu}_{1,T}, \bar{\mu}_{2,T})$. This, for example, will allow us to construct confidence intervals of $(\mu_1, \mu_2)$.

## 3 MAIN RESULTS

Before moving onto the details of the algorithm we introduce a class of bandit algorithms which we call *stable bandit algorithms*. Our first result, stated in Lemma 1, proves that stable algorithms ensures that the sample means $(\bar{\mu}_{1,T}, \bar{\mu}_{2,T})$ are asymptotically normal.

### 3.1 STABLE BANDIT ALGORITHMS

Throughout, we use $\mathcal{M}_T$ to denote a generic bandit algorithm with horizon $T$. Let $n_{a,t}(\mathcal{M}_T)$ denote the number of arm pulls of arm $a$ in $t$ rounds. We say an algorithm $\mathcal{M}_T$ is *stable* if for arms $a \in \{1, 2\}$ there exists *non-random* scalars $n_a^\star(\mathcal{M}_T)$ satisfying

$$(\texttt{stability:}) \qquad \frac{n_{a,T}(\mathcal{M}_T)}{n_a^\star(\mathcal{M}_T)} \xrightarrow{p} 1 \quad \text{for some} \quad n_a^\star(\mathcal{M}_T) \to \infty \quad \text{as} \quad T \to \infty. \qquad (1)$$

Here, the constants $\{n_a^\star(\mathcal{M}_T)\}_{a=1,2}$ above may depend on the parameters associated to reward distributions $\mathcal{P}_1$, $\mathcal{P}_2$ or other tuning parameters that are independent of the data collected using algorithm $\mathcal{M}_T$. Throughout, we hide the dependence of the algorithm $\mathcal{M}_T$ in $n_{a,T}$ and $n_a^\star$ for notational simplicity. Let us first prove a simple yet useful Lemma for stable algorithms:

**Lemma 1** *If an algorithm $\mathcal{M}_T$ is stable and the third moment of the arm-reward distribution $\mathcal{P}_a$ is bounded. Then for all arms $a \in \{1, 2\}$ the sample means are asymptotically normal. Concretely,*

$$\sqrt{n_{a,T}} \cdot (\bar{\mu}_{a,T} - \mu_a) \xrightarrow{p} \mathcal{N}(0, \sigma_a^2) \qquad (2)$$

**Proof of Lemma 1** Fix an arm $a$. Define the partial sum:

$$S_{a,t} = \sum_{\ell \leq t} (Y_\ell - \mu_a) \cdot \mathbf{1}_{\{A_\ell = a\}}$$

By construction, $S_{a,t}$ is a sum of Martingale difference sequence. Additionally, using the notation $\mathcal{F}_t := \sigma\{(Y_\ell, A_\ell)_{\ell \leq t}\}$ for the $\sigma$-field generated by data-set obtained up to stage $t$, we have

$$\sum_{1 \leq t \leq T} \text{Var}\left( \frac{1}{\sigma_a \cdot \sqrt{n_a^\star}} \cdot (Y_t - \mu_a) \cdot \mathbf{1}_{\{A_t=a\}} \mid \mathcal{F}_{t-1} \right) = \frac{n_{a,T}}{n_a^\star} \xrightarrow{p} 1.$$

In words, the sum of the conditional variances of the Martingale difference array stabilizes. Combining this with the assumption $n_a^\star \to \infty$ and using the fact that the third moment of the reward distribution is bounded (recall that rewards are sub-Gaussian) we see that the Lindeberg conditions of the Martingale Central Limit Theorem Dvoretzky (1972) are satisfied. Thus, applying the Martingale CLT from Dvoretzky (1972) we conclude

$$\sqrt{n_{a,T}} \cdot (\bar{\mu}_{a,T} - \mu_a) \xrightarrow{p} \mathcal{N}(0, \sigma_a^2) \qquad (3)$$

This completes the proof of Lemma 1.

**Confidence interval for** $\mu_a$**:**   Of course, we can estimate the reward variance by the sample variance estimate, and utilize Lemma 1 to construct confidence intervals for the unknown sample means $\mu_1$ and $\mu_2$. For instance, for any consistent estimator of $\widehat{\sigma}_{a,T}$ of $\sigma_a$, and given any target $\alpha \in (0,1)$ using Slutsky's theorem we conclude that

$$\lim_{T \to \infty} \mathbb{P} \left( \left[ \bar{\mu}_{a,T} - \widehat{\sigma}_{a,T} \cdot \frac{z_{\alpha/2}}{\sqrt{n_{a,T}}}, \bar{\mu}_{a,T} + \widehat{\sigma}_{a,T} \cdot \frac{z_{\alpha/2}}{\sqrt{n_{a,T}}} \right] \ni \mu_a \right) = 1 - \alpha.$$

See, the comments after Theorem 1 for a discussion on consistent estimator of $\sigma_a$.

## 3.2   Inference in 2-Batch bandits: Explore then commit (ETC) strategies

In this section, we focus on explore then commit-type strategies. Before doing so, we argue that many naive and intuitive algorithms are not stable.

### 3.2.1   Instability of vanilla-ETC strategy:

Arguably, the most naive and intuitive strategy is the explore then commit strategy which uses sample mean to decide which arm to commit to. Concretely, consider an $\epsilon$-greedy ETC algorithm where we

- Pull both arms with probability $1/2$ for a total of $2m$ times in the first batch.

- Define

$$(\epsilon\text{-greedy}) \qquad \hat{a}_{\max} = \arg \max_{a \in \{1,2\}} \bar{\mu}_{a,2m} \tag{4}$$

- For the remaining $T - 2m$ rounds, pull the arm $\hat{a}_{\max}$ with higher sample mean with probability $1 - \epsilon$, and the arm with lower mean with probability $\epsilon > 0$.

Let us discuss the stability property of the above algorithm. For simplicity, let us assume the reward distributions are Gaussian; i.e., $\mathcal{P}_1 \equiv \mathcal{N}(\mu_1, 1)$, $\mathcal{P}_2 \equiv \mathcal{N}(\mu_2, 1)$, and $m = \lceil T/4 \rceil$. In the case when the margin $\Delta = \mu_1 - \mu_2 = 0$, by symmetry we have

$$\mathbb{P}(\bar{\mu}_{1,2m} > \bar{\mu}_{2,2m}) = \mathbb{P}(\bar{\mu}_{2,2m} > \bar{\mu}_{1,2m}) = \frac{1}{2}$$

Thus for both arms $a \in \{1,2\}$ we have,   as $T \to \infty$

$$\frac{n_{a,T}}{T} \xrightarrow{p} \begin{cases} \frac{1}{4} + \frac{\epsilon}{2} & \text{with probability} \quad \frac{1}{2} \\ \frac{3}{4} - \frac{\epsilon}{2} & \text{with probability} \quad \frac{1}{2} \end{cases}$$

Stated differently, the $\epsilon$-greedy ETC algorithm from equation 4 is not stable when $\Delta = 0$. Invoking (Zhang et al., 2020, Theorem 6) we have the following lemma:

**Lemma 2** *(Zhang et al., 2020, Theorem 6) Suppose the data is collected using the ETC algorithm from equation 4. Then the sample mean for arm 1 is not asymptotically normal when $\Delta = \mu_1 - \mu_2 = 0$. In particular,*

$$\sqrt{n_{1,T}} \cdot (\bar{\mu}_{1,T} - \mu_1) \xrightarrow{d} Y \tag{5}$$

*where $Y = \sqrt{\frac{1}{3-\epsilon}} \left( Z_1 - \sqrt{2-\epsilon} Z_3 \right) \mathbb{I}_{Z_1 > Z_2} + \sqrt{\frac{1}{1+\epsilon}} \left( Z_1 - \sqrt{\epsilon} Z_3 \right) \mathbb{I}_{Z_1 < Z_2}$, and $Z_1, Z_2, Z_3$ are iid standard Gaussian random variables.*

This instability property of the $\epsilon$-greedy ETC algorithm also extends to other natural algorithms like $\epsilon$-greedy upper confidence bound (UCB) algorithm, and the non-normality of the sample means phenomenon still persists. See Appendix C of the paper Zhang et al. (2020) for more details.

### 3.2.2 A STABLE ETC-STRATEGY:

We are now ready to discuss a modification of the $\epsilon$-greedy ETC ( displayed in equation 4) which is stable. The algorithm proceeds in two stages:

- In the first stage, we select both arms $m$ times.
- At the end of first stage, we collect arms with high rewards and create an active set $\mathcal{A} \subseteq \{1,2\}$. In the second stage, we select all the arms in the active set $\mathcal{A}$ equally often.

The details of this two-batch method is detailed in Algorithm 1. We point out the strategy in Algorithm 1 draws motivation from elimination types algorithms studies Auer & Ortner (2010). We are now ready to analyze the stability of Algorithm 1.

---

**Algorithm 1** An Explore then Commit strategy

---

**Inputs:** Pair of integers $(T, m)$ with $1 \le m \le T/2$

**Batch 1**
Pull both arm $m$ times, and construct the active set after the total $2m$ arm-pulls

$$
\mathcal{A} := \left\{ a \ \mid \ \bar{\mu}_{a,2m} + \sqrt{\frac{2\log T}{n_{a,2m}}} \ge \max \left\{ \bar{\mu}_{1,2m} - \sqrt{\frac{2\log T}{n_{1,2m}}}, \bar{\mu}_{2,2m} - \sqrt{\frac{2\log T}{n_{2,2m}}} \right\} \right\} \tag{6}
$$

**Batch 2:**
**if** $T - 2m \ge 1$ **then**
    If the set $\mathcal{A}$ is singleton, pull the arm in $\mathcal{A}$ remaining $T - 2m$ times, or pull both arms with probability $1/2$, a total of $T - 2m$ times.
**end if**

---

**Condition on $m$:**    Let, $\Theta$ denote the collection of all problem dependent parameters. In Theorem 1, we allow any sequence of $m \equiv m(T, \Theta)$ that satisfies the following property:

$$
\frac{m \cdot \Delta^2}{8 \log T} \to \beta \quad \text{for some} \quad 0 \le \beta \le \infty \quad \text{as} \ T \to \infty. \tag{7}
$$

Here, the condition for $\beta = \infty$ means $\frac{m \cdot \Delta^2}{8 \log T} \to \infty$. The condition above, for instance allows for $m = T^\alpha$ for some $1 > \alpha > 0$, $m = \frac{2 \log T}{\Delta^2}$, or any constant value of $m$. Additionally, the condition 7 is always satisfied with $\beta = 0$ when $\Delta = 0$ for any value of $m$. Condition 7 rules out choices of $m$ — changing with $T$ — for which the ratio in 7 oscillates. The condition equation 7 allows for most choices of $m$ that are used in practice, especially when $\Delta$ is kept fixed as the number of $T$ increases. As we discuss later, the above condition also allows us to analyze the case when $\Delta$ is allowed to scale with the sample size $T$.

**Theorem 1** *Suppose $m$ satisfies condition equation 7 for some $0 \le \beta \le \infty$, and $T - 2m \to \infty$. Then Algorithm 1 is stable with the following choices of $n_1^\star, n_2^\star$*

$$
\text{If} \quad \beta \le 1, \quad n_1^\star = n_2^\star = \frac{T}{2} \tag{8}
$$

$$
\text{If} \quad \beta > 1, \quad \text{then} \quad n_1^\star = T - \frac{8\beta \log T}{\Delta^2} \quad \text{and} \quad n_2^\star = \frac{8\beta \log T}{\Delta^2} \tag{9}
$$

*Consequently, for both arms*

$$
\widehat{\sigma}_{a,T} \cdot \sqrt{n_{a,T}} \cdot (\bar{\mu}_{a,T} - \mu_a) \overset{d}{\to} \mathcal{N}(0,1).
$$

*Here, $\bar{\mu}_{a,T}$ denotes the sample mean, and $\widehat{\sigma}_{a,T}$ is any consistent estimator of variance $\sigma_a$.*

See Section A.2 for a proof of this theorem [1]. A few comments regarding Theorem 1 are in order.

---
[1]When $\beta = \infty$, we replace $\frac{8 \log T}{\Delta^2}$ by $m$ in equation 9.

**Estimating variance and statistical inference:** It turns out that under the assumptions of Theorem 1 sample variance estimate $\widehat{\sigma}_{a,T}$ is a consistent estimator of $\sigma_a$. Here,

$$\widehat{\sigma}_{a,T} = \frac{1}{n_{a,T}} \sum_{t=1}^{T} (Y_t - \bar{\mu}_{a,T})^2 \cdot \mathbf{1}_{A_t=a} \tag{10}$$

See Corollary 1 in the paper Khamaru & Zhang (2024) for a proof of consistency for $\widehat{\sigma}_{a,T}$. One can now easily create asymptotically exact $1 - \alpha$ confidence interval. In particular, given any $1 > \alpha > 0$ define the confidence interval $\mathcal{C}_{a,\alpha}$

$$\mathcal{C}_{a,\alpha} = \left[ \bar{\mu}_{a,T} - \widehat{\sigma}_{a,T} \cdot \frac{z_{1-\alpha/2}}{\sqrt{n_{a,T}}} \ , \ \bar{\mu}_{a,T} - \widehat{\sigma}_{a,T} \cdot \frac{z_{1-\alpha/2}}{\sqrt{n_{a,T}}} \right] \tag{11}$$

where $z_{1-\alpha/2}$ is the $(1 - \alpha/2)^{th}$ quantile of the standard Gaussian distribution. Then, we have that for both arms $a \in \{1, 2\}$ $\lim_{T \to \infty} \mathbb{P}(\mu_a \ni \mathcal{C}_{a,\alpha}) = 1 - \alpha$.

**Stability its and connections to Law of Iterated Logarithm:** It is interesting understand whether the bonus factor $\sqrt{2 \log T}$ plays any special role in stability, and whether we can replace it some other bonus factor. A careful look at the proof (see Section A.2) reveals that one can replace $\sqrt{2 \log T}$ in the equation 6 by any other bonus-term $q_T$ satisfying

$$\frac{\sqrt{2 \log \log T}}{q_T} \to 0 \quad \text{as} \ T \to \infty. \tag{12}$$

The term $\sqrt{2 \log \log T}$ above comes from the Law of Iterated Logarithm (LIL). In other words, the stability of Algorithm 1 is guaranteed as long as the bonus factor is $q_T$ *over-powers* the fluctuations in the sample means — which is governed by the Law of Iterated Logarithm. In such case, modifying the argument of Theorem 1 we obtain the following corollary:

**Corollary 1** *Suppose condition 12 holds, and $\frac{m\Delta^2}{4q_T} \to \beta$ for some $0 \le \beta \le \infty$. Then Algorithm 1 with bonus-term $q_T$ in place of $\sqrt{2 \log T}$ is stable. We have*

$$n_1^\star = n_2^\star = \frac{T}{2} \quad \textit{If} \quad \beta \le 1, \quad \textit{and}$$

$$n_1^\star = T - \frac{4\beta q_T^2}{\Delta^2} \quad \textit{and} \quad n_2^\star = \frac{4\beta q_T^2}{\Delta^2} \quad \textit{If} \quad \beta > 1.$$

### 3.2.3 DATA DEPENDENT STOPPING: OPTIMAL-REGRET WITH FREE INFERENCE

In Theorem 1 we assume that the choice of $m$ in Algorithm 1 is a pre-determined input to the algorithm, and it *does not depend* on the data collected by Algorithm 1. In this section, we analyze a two-stage algorithm where $m$ is dependent on the data, more formally a stopping time.

**Corollary 2** *Algorithm 2 is stable with*

$$\frac{n_{1,T}}{T - 4 \log T/\Delta^2} \xrightarrow{p} 1 \quad \textit{and} \quad \frac{n_{2,T}}{4 \log T/\Delta^2} \xrightarrow{p} 1$$

*Additionally, assuming $T\Delta^2 \ge 4e^2$, the regret $\mathbb{R}_T$ of Algorithm can be upper bounded by*

$$\mathbb{R}_T \le \frac{16 \log T}{\Delta} + \frac{120e\sqrt{\log(\Delta^2 T/4)} + 64e + 32}{\Delta} + 2\Delta$$

See Section A.3.1 for a proof of this corollary.

**Comparison with lower bound:** It is interesting to compare the regret of the Algorithm 2 with a lower bound for explore and commit-type algorithms. Following the work of (Garivier et al., 2016, Theorem 4) we have that for any uniformly efficient ETC strategies Lai & Robbins (1985); Garivier et al. (2016) we have that

$$\texttt{(Lower bound):} \qquad \lim_{T \to \infty} \inf \frac{\mathbb{R}_T}{\log T} \ge \frac{4}{\Delta}. \tag{13}$$

---
**Algorithm 2** ETC with stopping time
---
**Inputs:** Integer $T \geq 2$
Set $A_1 = 1$, $A_2 = 2$ and set $t = 2$
**while** $|\bar{\mu}_{1,t} - \bar{\mu}_{2,t}| \leq \sqrt{\frac{4 \log T}{(t/2)}}$ **do**
    Use $A_{t+1} = 1$ and $A_{t+2} = 2$, and set $t = t + 2$
**end while**

$$\mathcal{A} := \left\{ a \;\middle|\; \bar{\mu}_{a,t} + \sqrt{\frac{\log T}{(t/2)}} \geq \max\left\{ \bar{\mu}_{1,t} - \sqrt{\frac{\log T}{(t/2)}}, \bar{\mu}_{2,t} - \sqrt{\frac{\log T}{(t/2)}} \right\} \right\}$$

**if** $T - t \geq 1$ **then**
    If the set $\mathcal{A}$ is singleton, pull the arm in $\mathcal{A}$ remaining $T - t$ times, or pull both arms with probability $1/2$, a total of $T - t$ times.
**end if**
---

See Section 3 of work by Garivier et al. (2016) for more discussion on the lower bound. It is now interesting to understand the asymptotic behavior of the Algorithm 2. Assuming $\Delta$ is bounded by a constant, simple algebra yields

$$\lim_{T \to \infty} \sup \frac{\mathbb{R}_T}{\log T} \leq \frac{16}{\Delta}$$

Stated differently, Algorithm 2 ensures accurate asymptotic inference while matching the minimax-optimal regret up to a factor of 4.

### 3.3 INFERENCE IN B-BATCHED BANDITS

In this section, we focus on a batched bandit algorithm with $B$ batches. In each batch $1 \leq b \leq B$, we perform arm pulls a total of $2m$ times. Throughout this section, we assume $m$ is fixed, and we let the number of $B \to \infty$. We give details about our algorithm in Algorithm 3. Akin to the last section, we are interested in the stability of Algorithm 3.

We point-out that unlike Section 3.2.2 the number of arm pulls within each batch is *fixed*, i.e., not data-dependent.

---
**Algorithm 3** $B$-batch algorithm
---
**Input:** Pair of integer $(m, B)$ with $m, B \geq 1$.
Set $T = 2mB$, $\mathcal{A}_1 = \{1, 2\}$ and pull both arms $m$ times.
**for** $b = 1$ **to** $B - 1$ **do**
    Construct the active set

$$\mathcal{A}_{b+1} := \left\{ a \;\middle|\; \bar{\mu}_{a,2mb} + \sqrt{\frac{2 \log T}{n_{a,2mb}}} \geq \max\left\{ \bar{\mu}_{1,2mb} - \sqrt{\frac{2 \log T}{n_{1,2mb}}}, \bar{\mu}_{2,2mb} - \sqrt{\frac{2 \log T}{n_{2,2mb}}} \right\} \right\}$$

    If the set $\mathcal{A}_{b+1}$ is singleton, pull the arm in $\mathcal{A}_{b+1}$ $2m$ times, or pull both arms $m$ times.
**end for**
---

**Theorem 2** *Suppose $B \to \infty$, then Algorithm 3 is stable with*

$$n_1^\star = \frac{T}{2} \cdot \mathbf{1}_{\{\Delta=0\}} + \left( T - \frac{8 \log T}{\Delta^2} \right) \cdot \mathbf{1}_{\{\Delta>0\}} \qquad and$$

$$n_2^\star = \frac{T}{2} \cdot \mathbf{1}_{\{\Delta=0\}} + \frac{8 \log T}{\Delta^2} \cdot \mathbf{1}_{\{\Delta>0\}}. \tag{14}$$

*Consequently, for each arm $a \in \{1, 2\}$*

$$\widehat{\sigma}_{a,T} \cdot \sqrt{n_{a,T}} \cdot (\bar{\mu}_{a,T} - \mu_a) \xrightarrow{d} \mathcal{N}(0, 1).$$

*Here, $\bar{\mu}_{a,T}$ denotes the sample mean, and $\widehat{\sigma}_{a,T}$ is any consistent estimator of variance $\sigma_a$.*

See Section A.3 for a proof of this theorem. Just like our previous section, the results in Theorem 2 can be generalized to a general bonus term $q_T$ satisfying

$$\frac{\sqrt{2\log\log T}}{q_T} \to 0 \quad \text{as} \ \ T \to \infty.$$

In particular, for any $q_T$, the expression of $n_a^\star$ in equation 14 changes to

$$n_1^\star = \frac{T}{2} \cdot \mathbf{1}_{\{\Delta=0\}} + \left( T - \frac{4q_T^2}{\Delta^2} \right) \cdot \mathbf{1}_{\{\Delta>0\}} \qquad \text{and}$$

$$n_2^\star = \frac{T}{2} \cdot \mathbf{1}_{\{\Delta=0\}} + \frac{4q_T^2}{\Delta^2} \cdot \mathbf{1}_{\{\Delta>0\}}.$$

## 4 PROOFS

In this section, we prove our main Theorem 1, in part. Complete proof of all the results are deferred to the Appendix.

Define

$$g_T = \sqrt{2\log T} \quad \text{and} \quad h_T := \sqrt{7\log\log(4T) + 3\log 2} \tag{15}$$

$$\mathcal{E}_T := \left\{ |\bar{\mu}_{1,n_{1,t}} - \mu_1| \le \lambda_1 \frac{h_T}{\sqrt{n_{1,t}}} \quad \text{and} \quad |\bar{\mu}_{2,n_{2,t}} - \mu_2| \le \lambda_2 \frac{h_T}{\sqrt{n_{2,t}}} \ \forall \ t \in [T] \right\} \tag{16}$$

The proof utilizes the following lemma from (Khamaru & Zhang, 2024, Lemma 5.1). See also the work by Balsubramani (2014).

**Lemma 3** *Let* $X_1, X_2, \ldots$ *be i.i.d.* $\lambda_a$*-sub-Gaussian random variable with zero mean. Then the sample-mean* $\overline{X}_t := (X_2 + \ldots + X_t)/t$ *satisfies the following bound*

$$\mathbb{P}\left( \exists t \ge 1 : |\overline{X}_t| \ge \lambda_a \sqrt{\frac{9}{4t} \cdot \log \frac{(\log_2 4t)^2}{\delta}} \right) \le 2\delta$$

By assumptions the arm-means are sub-Gaussian with sub-Gaussian parameter bounded by 1.Thus, substituting $\delta = 1/\log(4T)$ in Lemma 3 and taking a union bound over both arms we obtain

$$\mathbb{P}\left( \mathcal{E}_T \right) = \mathbb{P}\left( |\bar{\mu}_{a,t} - \mu_a| \le h_T, \ \text{for all} \ \ 1 \le t \le T, \ \ 1 \le a \le 2 \right) \ge 1 - \frac{6}{\log(4T)} \tag{17}$$

### 4.1 PARTIAL PROOF OF THEOREM 1

Let us define two indicator variables

$$I_1 := \mathbf{1}_{\{1\in\mathcal{A}\}} \qquad \text{and} \qquad I_2 := \mathbf{1}_{\{2\in\mathcal{A}\}}.$$

From Algorithm 1 we have,

$$n_{1,T} = \begin{cases} T - m & \text{if } I_2 = 0, \\ m & \text{if } I_1 = 0, \\ m + \Sigma_{i=1}^{T-2m} V_i & \text{if } I_1 = I_2 = 1. \end{cases} \tag{18}$$

where $V_i \sim \text{Bern}\left(0, \frac{1}{2}\right)$ for $1 \le i \le T - 2m$

The proof follows by analyzing the random variable $n_{1,T}$ under the high probability event $\mathcal{E}_T$. Using the bound equation 16, we get that $\mathbb{P}(\mathcal{E}_T) \ge 1 - \frac{6}{\log 4T}$. Thus, it suffices to study the behavior of $n_{1,T}$ on the high-probability event $\mathcal{E}_T$.

CASE 1: $\beta \le 1$:

We have that for large $T$s

$$\{I_2 = 0\} \cap \mathcal{E}_T = \left\{\bar{\mu}_{2,2m} + \frac{g_T}{\sqrt{n_{2,2m}}} < \bar{\mu}_{1,2m} - \frac{g_T}{\sqrt{n_{1,2m}}}\right\} \cap \mathcal{E}_T$$

$$\overset{(i)}{\subseteq} \left\{\mu_2 - \lambda_2 \frac{h_T}{\sqrt{n_{2,2m}}} + \frac{g_T}{\sqrt{n_{2,2m}}} < \mu_1 + \lambda_1 \frac{h_T}{\sqrt{n_{1,2m}}} - \frac{g_T}{\sqrt{n_{1,2m}}}\right\}$$

$$= \left\{\mu_2 - \lambda_2 \frac{h_T}{\sqrt{m}} + \frac{g_T}{\sqrt{m}} < \mu_1 + \lambda_1 \frac{h_T}{\sqrt{m}} - \frac{g_T}{\sqrt{m}}\right\}$$

$$= \left\{\frac{1}{\sqrt{m}}(-(\lambda_1 + \lambda_2)h_T + 2g_T) < \Delta\right\}$$

The step (i) uses the property of the event $\mathcal{E}_T$. Now note that the set in the last line is empty when $\Delta = 0$ for large $T$; this is because $\frac{g_T}{h_T} \to 0$ as $T \to \infty$. When $\Delta > 0$ with $\beta \le 1$, we have using condition equation 7 and $h_T/g_T \to 0$ we have

$$\frac{1}{\Delta\sqrt{m}}(-(\lambda_1 + \lambda_2)h_T + 2g_T) \to 1/\sqrt{\beta} \ge 1.$$

where we have used the notation $1/0 \equiv \infty$ for $\beta = 0$, and we have $\mathbb{P}(\mathcal{E}_T \cap \{I_2 = 0\}) \to 0$ when $0 \le \beta \le 1$. Since $\mu_1 \ge \mu_2$ by assumption, it is immediate to verify that $\mathbb{P}(\mathcal{E}_T \cap \{I_1 = 0\}) \to 0$. Thus we have

$$\mathbb{P}(\{I_1 = 1\} \cap \{I_2 = 1\} \cap \mathcal{E}_T) \to 1, \tag{19}$$

When $I_1 = I_2 = 1$, we have $n_{1,T} = m + \sum_{i \le T-2m} V_i$, and we have

$$\frac{m + \sum_{i \le T-2m} V_i}{T/2} = 1 + \frac{\sum_{i \le T-2m}(V_i - \frac{1}{2})}{\frac{T}{2}} \xrightarrow{p} 1$$

where the last deduction above uses $T - 2m \to \infty$ and the weak law of large numbers. Using a similar argument for $n_{1,T}$, and putting together the pieces we conclude

$$\frac{n_{1,T}}{T/2} \xrightarrow{p} 1 \qquad \text{and} \qquad \frac{n_{2,T}}{T/2} \xrightarrow{p} 1.$$

The proof of the case $\beta > 1$ is similar, and the details are moved to Appendix .

## 5    DISCUSSION

In this paper, we discussed the problem of statistical inference when data is collected using a batched bandit algorithm. We introduced the concept of stable bandit algorithms, which allows for straightforward statistical inference even when the dataset is not i.i.d. For instance, the sample arm means are asymptotically normal when data is collected using a stable bandit algorithm. We also argue that such stable algorithms do not sacrifice regret and are optimal up to a constant factor in certain cases. Our work bridges the gap between regret minimization and statistical inference, two historically conflicting paradigms in sequential learning environments.

While we focused on two-armed bandit problems in this paper, several interesting questions remain. For instance, it would be interesting to extend our results to the $K$-armed case. In our $B$ batched Algorithm 3, the number of arm-pulls ($m$) in each batch is kept fixed. It would be interesting to understand the stability properties of our algorithm when the number of arm-pulls are allowed to grow with $T$.

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
