# OpenReview forum: "Stable batched bandit:  Optimal regret with free inference"
_ICLR.cc/2025/Conference — Submitted to ICLR 2025_

### Official Review · Reviewer_Ptna · 2024-10-28

**Soundness:** 3
**Presentation:** 2
**Contribution:** 2
**Rating:** 5
**Confidence:** 3

**Summary:**

Given a set of arms and a bandit algorithm that collects data, this paper aims to understand the asymptotic properties of the empirical mean of each arm observed by running the algorithm. The focus is on the setting of 2 arms. While existing methods do not typically use information about the algorithm itself, this paper takes an algorithm-dependent approach, via the notion of stability (defined in Section 3.1). In particular, for stable algorithms, the empirical means are asymptotically normal (Lemma 1). This means that one can readily construct confidence intervals for the true means.

The paper discusses two types of algorithms: 2-batch and $B$-batch. For the former, they show that a vanilla $\epsilon$-greedy Explore Then Commit (ETC) strategy is not always stable (Lemma 2). They then modify the algorithm into a stable ETC variant (Algorithm 1); its stability is proved in Theorem 1. In both of these algorithms, the learner explores for a fixed number $\sim m$ of rounds and subsequently acts greedily. The authors then devise Algorithm 2, where they replace $m$ with a random stopping time. They then show that Algorithm 2 is stable (Corollary 2) and has asymptotically optimal regret, up to constants.

Lastly, they study a $B$-batch procedure. Algorithm 3 pulls each arm $\sim m$ times in each one of $B$ batches. Theorem 2 then shows that this algorithm is stable (when $m$ is fixed and $B\to\infty$).

**Strengths:**

- Overall, the paper is well-structured and clear, but would benefit from some clarifications (addressed below).

- It is an interesting approach to show asymptotic normality based on properties of the algorithm.

- I think asymptotic normality is indeed important to be able to do inference on the arms (e.g., construct confidence intervals).

- The algorithms are relatively clear and simple to understand.

**Weaknesses:**

- The 2 arm assumption is fairly limited. It might be productive to discuss how this can be extended.

- The paper would benefit from additional intuition on some concepts, such as (i) the definition of stability and (ii) the ratio in the constraint on $m$ of Equation (7).

- In Section 3.1, the authors should define $z_\alpha$.

- In Corollary 2, the authors should define regret.

- In Theorems 1 and 2, in the convergence in distribution to the Gaussian, shouldn't the LHS be dividing by $\hat\sigma_{a,T}$ instead of multiplying?

**Questions:**

- Is there a reason why in Algorithm 3, provided $\mathcal{A}_{b}$ contains both arms, we pull each one exactly $m$ times as opposed to the other algorithms, where we pull each one with probability $1/2$?

- How do you think the analysis for $K$ arms would work? Would it be a simple extension, or require more sophisticated techniques?

---

> ### Author Response · Authors · 2024-11-19
>
> We thank the reviewer for their valuable comments. Below, we address the reviewer's main points.
>
>
> Question:How do you think the analysis for K arms would work? Would it be a simple extension, or require more sophisticated techniques?
>
> Answer: Due to space constraints, we provide the answer to this question as a general response to all the reviewers.
>
> Question:
> In Theorems 1 and 2, in the convergence in distribution to the Gaussian, shouldn't the LHS be dividing by σ_a,T  instead of multiplying?
>
> Answer: Thanks for pointing this  out. You are right in pointing this out. We will fix it in the final version.
>
> Question:
> What is intuition on some concepts, such as (i) the definition of stability and (ii) the ratio in the constraint on $m$ of Equation (7).
>
> Answer: First note that, if we pull arm 1 with probability p and arm 2 with probability 1 - p at every round with 0< p < 1, then by central limit theorem, the sample  means of each arm are asymptotically normal. In the bandit setting, let p_t = P(selecting arm 1 at round t | all data up to time t - 1). Then a sufficient condition for stability is p_t   —> p (in probability). This is essentially the same (in a limiting sense) as the iid set up where we pull arm 1 with probability p at every round. The condition of stability is a much more general version of this intuition, where p_t’s need not converge, they can even oscillate, and we only require that the average of p_t’s converge.
>
>
> The intuition of $m$ of equations (7) - (9):
>
> It is helpful to rewrite conditions (7) - (9) as:
>
> m > 8logT / \delta^2   and m \leq 8logT / \delta^2.
>
> Theorem 1 states that when m > 8logT / \delta^2, stage 1 of Algorithm 1 can successfully detect mu_2 < mu_1. When m  \leq  8logT / \delta^2, the stage 1 of the algorithm cannot distinguish between the \mu_1 and _mu_2 = \mu_1 - \delta. We point out that the sample size 8logT / \delta^2 is not the optimal number of samples that are needed (the Log-term can be improved), and this sample-size requirement is specific to this algorithm which maintains stability of the ETC algorithm we analyze.
>
> Question:
> Is there a reason why in Algorithm 3, provided it contains both arms, we pull each one exactly times as opposed to the other algorithms, where we pull each one with probability 0.5?
>
> Answer: We thank the reviewers for pointing this out. Our current proof becomes simple if we choose all active arms an equal number of times (instead of pulling them with equal probability). However, we agree with the reviewer that this is another natural alternative, and we are working on a proof. In the final version of the paper, we intend to add some numerical experiments with the modified version of Algorithm 3 to check (at least numerically) whether it is stable or not.

---

> > ### Comment · Reviewer_Ptna · 2024-11-26
> >
> > I appreciate the authors' detailed responses. The extension of the $B$-batch algorithm to $K$ arms seems like a natural one. I also like the intuition presented on the stability definition and think the paper would benefit from having something along those lines in the main text. Nevertheless, I agree with the other reviewers that the topic lacks motivation. It would be interesting to see a more comprehensive study on the tradeoffs between CLT-type results and regret guarantees. Hence, I would like to keep my score.

---

### Official Review · Reviewer_dcVu · 2024-10-31

**Soundness:** 3
**Presentation:** 3
**Contribution:** 2
**Rating:** 5
**Confidence:** 4

**Summary:**

The paper considers the problem of statistical inference in context of sequentially / adaptively collected (and hence non-i.i.d.) data as arising in the multi-armed (batched) bandit setting.  In particular, the authors introduce a notion of stability for bandit algorithms, and prove that under their stability condition, sample means for arms are asymptotically normal, which allows the construction of valid confidence intervals.  They show that a commonly considered, simple explore-then-commit (ETC) strategy is unstable (and indeed yields non-normal reward means). They then provide a stable alternative, which, in the first stage, samples arms equally often, and in the second stage samples all plausibly optimal arms equally often. They also provide variants of this scheme, with adaptively selected duration of the first stage (which yields an algorithm with optimal asymptotic minimax regret up to a constant of 4), or with multiple stages of fixed length.

**Strengths:**

I very much appreciate the perspective for analysing multi-armed bandits using the lens of stability. While stability has been known to imply generalisation bounds in classical learning theory, its application to the analysis to bandits I had not seen before.  I could imagine it becoming a fruitful direction for future research in this area.  The fact that stability allows treating the data as effectively generated i.i.d. is potentially impactful.  The paper is generally clearly written (albeit there’s a number of typos).

**Weaknesses:**

While very interesting conceptually, the submission only considers a rather restricted setting:  A variant of batched bandits with two arms only.  While in Section 2, the authors claim that “many” of their results generalise to the K-armed setting, no further information is provided (which results? How do the results generalise?), and in Section 5, they mention the extension to the K-armed case as interesting future work.  In my opinion, stability would become a much more interesting concept if the results could be generalised to richer families of bandit instances, e.g., bandits with structured reward functions (e.g., linear bandits).  Besides the limitation to two arms, all presented results are only asymptotic in nature.  Thus, the work appears a bit preliminary for me.

Another, and perhaps more severe, concern is with the novelty of the presented framework.  The authors cite a paper entitled “Inference with ucb”.  Aiming to understand the relative contributions of the present submission, I found an article with the title “Inference with the upper confidence bound algorithm” on arXiv (https://arxiv.org/pdf/2408.04595).  This paper appears to introduce the same notion of stability for bandit algorithms and uses it to analyse the upper-confidence bound (ucb) bandit algorithm (in fact, Lemma 1 appears to be shown in that paper as well).   The paper appears to provide the same general conclusion (i.e., under stability, data can be effectively treated as i.i.d.) as in the ICLR submission.  The main difference appears to be that an analysis is presented for a different family of bandit algorithms (ucb).  With respect to this paper, the ICLR submission looks rather incremental, even more so in light of the rather restricted setting (as argued above).

The manuscript has a number of typos (here’s some of them):
- 60: maybe -> may be
- 158: genrality
- 224: on consistent estimator of
- 286: pull both arm

**Questions:**

- Can you please clarify, in your view, the relation and relative contribution to the paper https://arxiv.org/pdf/2408.04595?
- Can you please elaborate on whether there are meaningful connections to generalisation bounds obtained from stability in classical learning theory [cf. A,B]?
- Section 2 makes the claim that “many” results generalise to the K-armed setting.  Which results?  And how do they generalise?
- \sigma_{a,T} as used, e.g., in line (10) seems to refer to the variance, in (11) it seems to be used as standard deviation.  Can you please clarify?

---

> ### Author Response · Authors · 2024-11-19
>
> We thank the reviewer for their valuable comments. Below, we address the reviewer's main points.
>
> Question:
> Can you please clarify, in your view, the relation and relative contribution to the paper by Khamaru and Zhang?
>
> Answer:
>  Due to space constraints, we provide the answer to this question as a general response to all the reviewers.
>
> Question:
> Section 2 makes the claim that “many” results generalize to the K-armed setting. Which results? And how do they generalize?
>
> Answer:
>
> Theorem 3 in our paper has a natural K-arm generalization. We can show that for a K-arm bandit problem with B-batches, the elimination algorithm remains stable for K-arms, and we can quantify the limiting distribution of all K-arms. The extension to K-arms for the Batched Bandits can be implemented as follows:
>
> At each round b, we maintain a set of active arms A_b – the set of arms to be pulled in that round. A_b is defined as: A_b = {a ∈ A_{b-1}: UCB_of_arm_a > max_{a' ∈ A_{b-1}}[LCB of arm a']}
>
> In other words, we compare the Upper Confidence Bound of each arm with the maximum Lower Confidence Bound among arms in the previous active set A_{b-1}. An arm remains in the new active set A_b if its UCB exceeds this maximum LCB. We can prove this algorithm is stable, and consequently, all K arm-means are asymptotically normal. We will include this result in the final version of the paper. Please let us know if you would like us to provide additional details on the proof techniques or main results for the K-armed case in the review response.
>
>
> Question:
> Can you please elaborate on whether there are meaningful connections to generalization bounds obtained from stability in classical learning theory [cf. A,B]?
>
> Answer: Could you please provide detailed references to the paper [A,B]. We could not find these two references in the review.
>
> Question
> \sigma_{a,T} as used, e.g., in line (10) seems to refer to the variance, in (11) it seems to be used as standard deviation. Can you please clarify?
>
> Answer: Thank you for catching the typo. In equation (10) it should be \sigma_{a,T}^2.

---

> > ### Comment · Reviewer_dcVu · 2024-11-19
> > **Missing references**
> >
> > Thank you for your response.  I apologize for the missing references, which are:
> >
> > [A] O. Bousquet, A. Eliesseff, Stability and Generalization. JMLR 2002
> >
> > [B] M. Hardt, B. Recht, Y. Singer, Train faster, generalize better: Stability of stochastic gradient descent. ICML 2016

---

> ### Author Response · Authors · 2024-11-20
>
> Thank you for providing the references. These two stability definitions are different. The notion of "stability" in papers [A] and [B] refers to the stability of an algorithm/estimator - specifically, how much the estimator changes when part of the training data is swapped. In this case, the data is assumed to be i.i.d.
>
> On the contrary, the notion of stability used in the current work refers to a property of the data-collection procedure itself; it captures how similar sequential data collection is to i.i.d. data.

---

> > ### Comment · Reviewer_dcVu · 2024-11-25
> >
> > Thank you for the response.  I agree that the settings and definitions of stability are different.  I was rather interested whether there is any meaningful technical connection. In any case, I would like to keep my score and recommendation.

---

### Official Review · Reviewer_52RF · 2024-11-02

**Soundness:** 3
**Presentation:** 3
**Contribution:** 2
**Rating:** 5
**Confidence:** 4

**Summary:**

The authors introduce a class of algorithms known as "stable bandit algorithms" in which classical statistical methods used for iid data can be used for inference. Most of the results and discussion in the main paper is for 2 arm bandit setting, and this case, the authors show that the CLT holds for the sample means of the arms. The authors also show that the vanilla epsilon greedy explore-then-commit algorithm does not satisfy the stable bandit setting, and further go on to show that the sample means of the arms arms in this strategy does not satisfy the CLT. As a subsequent step, the authors propose a modified version of the algorithm that does belong to the class of stable bandit algorithms, and prove that its regret is asymptotically optimal (upto a constant factor). The authors also introduce another another algorithm "B-batch algorithm" that is also shown to be stable, and have nice asymptotic properties of the sample means of the arms.

**Strengths:**

In my opinion, the strength of the paper is how simple and elegant the theory is. It's nice to see that simple bandit algorithms that require such little computations have nice statistical properties.

**Weaknesses:**

1. I do not fully understand the motivation of the results. The authors claim "bridging gap between statistical inference and minimizing regret" but I do not understand what they mean by that exactly -- can the authors provide more concrete examples on how they do this? They discuss how previous analysis use "Martingale structure" in sequentially collected data for analysis, but so what?

2. I also don't understand what these results lead to in the practical / theoretical sense. The only future work the authors discuss is extension to K arm bandit setting: Can the authors provide specific examples of practical applications or theoretical implications of their results. How can stable bandit algorithms be applied in real-world scenarios?

3. The authors mention "many" of the results in the paper extend to K-arm bandit settings, but which ones? I did not see any discussion on this topic -- the authors should let us know how the results extend to K-arm settings, and also the implications.

4. I don't know why these results are complex / novel -- could the authors explain why this analysis is special in the context of bandit literature?

Overall in my opinion, I don't think the paper is appropriate for the venue. Simple, elegant results but it is not motivated enough either in terms of theory or practice.

**Questions:**

See above.

---

> ### Author Response · Authors · 2024-11-19
>
> We thank the reviewer for their valuable comments. Below, we address the reviewer's main points.
>
> Question 1:I do not fully understand the motivation of the results. The authors claim "bridging the gap between statistical inference and minimizing regret" but I do not understand what they mean by that exactly -- can the authors provide more concrete examples on how they do this?
>
>
> Answer:
> We wanted to argue that regret minimizing algorithms are not always friendly to statistical inference; in particular, it might be hard to determine the distribution of the sample arm means. One can of course pull each arm with constant proportion at every round (randomized control trial) which always ensures that the sample arm means are asymptotically normal, but this algorithm yields large regret of order O(T). Put simply, the criteria for easy statistical inference using normality and regret minimization seem to be two competing criteria, and it is not clear whether both conditions can be achieved by a single algorithm.
> The main contribution of our paper is that for some bandit algorithms, namely stable (Lai-Wei 1982) bandit algorithms with low O(√T) regret, the asymptotic normality of sample means is not broken. Lemma 2 of our paper also provides examples of natural bandit algorithms where this stability condition as well as the asymptotic normality condition is broken.
>
>
> Question 2:
> They discuss how previous analysis uses "Martingale structure" in sequentially collected data for analysis, but so what?
>
>
> Answer:
> The predominant technique for constructing confidence intervals is the techniques of self-normalized Martingales by Abbasi et. al. Such analyzes apply to “any” bandit algorithms, and do not use any property of the specific bandit algorithm that is used to collect. On the contrary, our asymptotic normality results only apply to specific bandit algorithms which are stable, and  it is straightforward to see that the resulting confidence intervals are sharper than the existing CI’s obtained from concentration inequalities.  See for instance, Lemma 2 for examples of natural algorithms which are unstable.
>
>
> Question 3:
> I also don't understand what these results lead to in the practical / theoretical sense. The only future work the authors discuss is extension to K arm bandit setting: Can the authors provide specific examples of practical applications or theoretical implications of their results. How can stable bandit algorithms be applied in real-world scenarios? The authors mention "many" of the results in the paper extend to K-arm bandit settings, but which ones? I did not see any discussion on this topic -- the authors should let us know how the results extend to K-arm settings, and also the implications.
>
>
> Answer: The practical implication of our theory is to use asymptotic normality based confidence intervals for bandit data. We note that the data collected in a bandit algorithm is non-iid in nature, and it is not clear why, even in the simple case of multi-armed bandits, the sample means will be asymptotically normal. Indeed, as we argued in Lemma 2 of the paper, for many standard bandit algorithms — including the popular UCB algorithm — the sample mean is not asymptotically normal for 2-armed batched bandit problem with number of batches = 2. The main contribution of our paper is to give examples of algorithms, in a batched bandit setting, where the sample means of the arms are asymptotically normal. We also refer the reviewer to the reply to Reviewer 2’s questions for more details on the contribution of our work compared to previous works.
>
> Theorem 3 in our paper has natural K-arm generalization. We can show that if we consider a K-arm bandit problem with B-batches, then the elimination algorithm is stable for K-arms, and we can quantify the limiting distribution of the K-arms. The extension to K arm for the Batched Bandits can be done as follows:  At each round, say b, we keep a set of active arms A_b – a set of arms that will be pulled in round b. A_b = \{ a \in A_{b - 1} : UCB_of_arm_a > max_{a’ \in A_{b - 1}} [ LCB of arm a’] \}. In other words,  we compare the Upper Confidence bound of a particular arm with the maximum lower confidence bound of the arms remaining in the previous active set A_{b - 1}, and keep all the arms in the new active set A_b for which UCB is large. We can show this  algorithm is stable, consequently, all K arm-means are asymptotically normal. We intend to add this result in the final version of the paper.
>
>
> For more clarity check the general response to all reviewers.

---

### Official Review · Reviewer_Emtd · 2024-11-02

**Soundness:** 2
**Presentation:** 2
**Contribution:** 2
**Rating:** 3
**Confidence:** 4

**Summary:**

This work explores statistical inference under sequentially and adaptively collected data. The authors focus on the batched bandit setting and show that ETC type policies are poor with inference due to lack of a property called "stability". They show that a form of Successive Elimination algorithm achieves stability and have good asymptotic inference properties as well as optimal regret bounds. They also generalize their results from the 2-batched setting to a multi-batched setting.

**Strengths:**

The context is relatively easy to follow. The problem considered is interesting.

**Weaknesses:**

Though this paper investigates an interesting and important problem, I am afraid the preprint is far from being ready for publish.

1. Contribution. What I am very confused is you contribution.
- You mention in the abstract you show that popular algorithms including the greedy-UCB algorithm and $\epsilon$-greedy ETC algorithms are not stable. Maybe I missed it, but where is it in the main context? Seems you only show the instability of the simple $\epsilon$-greedy ETC. The short argument on Line 268-269 seems to be very vague and built entirely on Zhang et al. (2020).
- You mention it is possible to minimize regret without sacrificing the ease of performing statistical inference. I think this is an overclaim. Your regret rate is in a non-asymptotic sense, but your inference task is in an asymptotic sense. Convergence in probability is a relatively weak measure --- it is a point-to-point convergence and one does not know the convergence rate. Also, in your result you show a sacrifice of factor $4$ in the regret bound.
- More importantly, it is very unclear how your work should be placed compared to Khamaru & Zhang (2024) which you also cited in the proof section. Khamaru & Zhang (2024) revisited the stability property, gave a detailed investigation of the asymptotic statistical properties of standard UCB. They also investigated the multi-armed setting (the number of arms $K$ can even be scaling with $T$).
  - Can you elaborate on your additional contribution? Stability is not new. The algorithms (ETC and Elimination) are not new. The results seems to be similar (or maybe even weaker) compared to those from Khamaru & Zhang (2024).
  - Also, it seems the technical tools are largely following the literature. Is there any intrinsic difficulty within (a) studying Elimination rather than UCB and (b) studying the batched setting?

2. Writing. The writing of this paper is relatively poor.
- The results and proofs are written in a rather arbitrary way. Some gave a partial proof with equations, some invoked other works directly, while some simply mentioned that "a careful look at the proof reveals that one can replace ...". The content is indeed easy to follow, but the reading experience is not good.
- In Theorem 1 you are taking $m$ and $T$ to $+\infty$ such that $T-2m$ goes to $+\infty$. But in Theorem 2 the scaling of $m$ is unclear. Seems there is no? There are also many typos in the paper. For example, in both Theorem 1 and 2, the empirical variance term is in the wrong place; in Line 796-797, the equation after "the fact" is strange and contents after "Recall" are not in math format; the usage of ∴ is not formal.
- There is only a small simulation example at the beginning of the paper. I understand the paper is focusing on theory, but since you claim you propose new algorithms (which are not hard to implement), it is anticipated that more comprehensive experiments should be conducted.

I believe the authors should provide convincing response to my concerns above (particularly the contribution part), in case there is a small possibility that I misunderstood the paper.

**Questions:**

See Weaknesses.

**Details Of Ethics Concerns:**

The definition of stable algorithms and the instability of $\epsilon$-greedy have also appeared in Khamaru & Zhang (2024). They cited the paper in the proof section but not in literature review or the contribution section.

---

> ### Author Response · Authors · 2024-11-19
>
> We thank the reviewer for their valuable comments. Below, we address the reviewer's main points.
>
> Question 1: What I am very confused is your contribution. More importantly, it is very unclear how your work should be placed compared to Khamaru & Zhang (2024) which you also cited in the proof section. Khamaru & Zhang (2024) revisited the stability property, gave a detailed investigation of the asymptotic statistical properties of standard UCB.
>
>
> Answer:
> Due to space constraints, we provide the answer to this question as a general response to all the reviewers.
>
>
> Question 2:
> You mention in the abstract you show that popular algorithms including the greedy-UCB algorithm and -greedy ETC algorithms are not stable. Maybe I missed it, but where is it in the main context? Seems you only show the instability of the simple -greedy ETC. The short argument on Line 268-269 seems to be very vague and built entirely on Zhang et al. (2020).
>
> Answer:
> The results that both greedy UCB and greedy-ETC is unstable follows from Zhang et al. 2020. Also see the work by Bibaut and Kallus 2024 arXiv preprint arXiv:2405.01281 (section 3). This is not our contribution, and we briefly mention one of such result to answer a natural question: why not UCB in batched bandit algorithms in two batches. We are happy to provide a detailed proof if it is helpful.
>
> Question 3:
> You mention it is possible to minimize regret without sacrificing the ease of performing statistical inference. I think this is an over-claim.  Also, in your result you show a sacrifice of factors in the regret bound.
>
> Answer:
> We thank the reviewer for this question. All we wanted to argue in Theorem 2  is that while looking for “stable algorithms” we do not lose much in terms of regret. This question is natural because the notion of stability is not very helpful if the regret of stable algorithms are much worse than the minimax optimal or asymptotically optimal regret (Lai Robbins). In Theorem 2 we argue that the asymptotic regret of of Algorithm 2 is at max 4 times the asymptotically optimal regret, and consequently, the minimax regret of this algorithm is  \sqrt{TlogT} — which is sub-optimal up to a factor \sqrt{logT}.  Put simply, we get inference for free without sacrificing the regret by too much.
>
> Question 4:
> Your regret rate is in a non-asymptotic sense, but your inference task is in an asymptotic sense. Convergence in probability is a relatively weak measure --- it is a point-to-point convergence and one does not know the convergence rate.
>
> Answer:
> As we mentioned in Lemma 1 [Lai and Wei Theorem 3], the notion of stability — which only requires convergence of n_{a,T}/n_a^\star to 1 -- is sufficient to get asymptotic normality of the sample mean. Hence, the weaker the necessary condition, the better. We agree that the rate of convergence is not clear from our calculations. This question, while very interesting, is beyond the current scope of the paper. However we can always verify the numerical performance (empirical coverage) of the confidence interval, and the asymptotic distribution of the sample mean.  Our appendix section details some numerical simulations which show that asymptotic distribution of the sample means are indeed in good accordance with a normal distribution. We will include additional simulations detailing the coverage of the confidence intervals in the final version of the paper.

---

> > ### Comment · Reviewer_Emtd · 2024-11-25
> >
> > I would like to thank the authors for the response which has partially addressed my concerns. I have removed the flag for ethics review, but still I think the current version lacks enough comparison to existing works, particularly Khamaru & Zhang (2024) and Zhang et al. (2020). It seems to me now that the main contribution mainly lies in the interesting finding that Successive Elimination can work in a few-batched bandit setting. I have to say the contribution part in the paper still seems to be a bit inflated to me, since the first bullet is largely originating from the literature. For the third bullet, it is unclear in Section 3.3 Theorem 2 why $B$ is taken to infinity --- isn't this making the whole problem similar to standard bandit setting with no batch constraints, deviating from your main focus? Apart from these, the current version lacks enough depth and clearness from different perspectives (also suggested by other reviewers, e.g., asymptotic nature of CLT, technical novelty, restricted setting, presentation clarity).
> >
> > In conclusion, I admitted that the finding is interesting, but I decided to maintain my score and I hope the authors can explore more on the topic and derive more results. I am willing to discuss further if there are any questions.

---

### Official Review · Reviewer_qeHo · 2024-11-04

**Soundness:** 4
**Presentation:** 2
**Contribution:** 2
**Rating:** 3
**Confidence:** 4

**Summary:**

Summary:
The problem studied is to design multi-armed bandit algorithms for stochastic bandit problems
so that the central limit theorem (CLT) holds for the rewards collected for each of the arms in the limit of infinitely many interactions. Let's call a bandit algorithm CLT friendly if it satisfies this criterion.

The notion of "stability" of bandit algorithms is introduced. According to this definition, a bandit algorithm is stable, if for any bandit instance of interest, there exist deterministic sequences $\{ n_{a,t} \}$ such that $\frac{N_{a,t}}{n_{a,t}}$ converges to one in probability where $N_{a,t}$ is the number of pulls of arm $a$ up to round $t$ on the given instance (a random quantity).
A simple calculation shows that stable bandit algorithms are CLT friendly.
The authors also show that ETC (explore for, say, half the time, then choose the better arm out of, say, two) is not stable. They also cite previous research that shows that ETC is not "CLT friendly", which indicates that enforcing stability may be necessary for CLT friendliness.
Then they design an ETC-style method, which explores in the first phase, but then instead of choosing greedily, uses confidence bounds with the data of the first phase to eliminate arms. If multiple arms remain, the algorithm splits the remaining time equally between them (the authors use randomization for this). If a single arm remains, that arm is pulled up to the end. This algorithm is shown to be stable.

Significance: The problem is not entirely new, several authors looked into CLT friendliness previously. This reviewer is not completely sold on this notion: CLT is truly asymptotic, it is unclear what this notion really buys for practice if anything. Also CLT friendliness could be achieved easily if we did not demand to use all rewards from all the arms: just allocate a fixed, even diminishing portion, of all time steps to uniformly exploring the arms and use the rewards collected during this period. The results of this paper are not strong enough to discard an algorithm like this on reasonable grounds. In other words, not much depth is achieved in the paper.

Novelty: The notion of stability is novel. The proofs are quite standard/automatic (even though I am not completely happy with how, e.g., the proof of Theorem 1 is done).

Related work: Somehow the authors want to connect this to batched bandits, but at least the main paper did not do much with this. The algorithm presented for the batched case with B cases raises more questions than it answers (maybe a presentation issue).

Soundness: I think the claims made are correct. I verified things to a reasonable degree in the main text.

Presentation:  There are a number of typos, grammatical issues (e.g., line 155: "We assume Let ...", line 158: genrality, etc.) I will list a few more of these at the end. In the algorithms, the authors use "or", but this should be "otherwise" (last line of all algorithms). This was very confusing. Also, the proof of Theorem 1 is quite messy (one of the two proofs in the main body). The authors state "It suffices to study the behavior of $n_{1,T}$ on the high probability event $\mathcal{E}_T$". Why? In what sense? (Later we find out, but this is not the sign of a well written text.) Also, only in the middle of the proof we find out that there will be two cases based on the value of $\beta$. This proof definitely can use polishing, as can the rest of the paper.

**Strengths:**

The paper does make novel contributions.

**Weaknesses:**

I did not find the topic well motivated. The paper feels weak on contributions: A strong paper would study the tradeoff between CLT friendliness and performance; nothing of this form is attempted here. The results, while they appear to be correct, do not require much effort. The batch bandit version of the algorithm is, hmm, unexpected. The presentation is poor; it feels that the paper need much work.

**Questions:**

It seems that in the batch bandit version of the algorithm in every batch, only data for that batch is used. The effect of previous batches is discarded. At least this is how the algorithm seems to be defined. While this may make the method CLT friendly, this will be a very bad bandit method. What is the point here?

---

> ### Author Response · Authors · 2024-11-19
>
> We thank the reviewer for their valuable feedback. We will revise the paper to improve clarity and readability of the proofs. Below, we address the reviewer's main points.
>
>
> Question 1:
> It seems that in the batch bandit version of the algorithm in every batch, only data for that batch is used. The effect of previous batches is discarded. At least this is how the algorithm seems to be defined. While this may make the method CLT friendly, this will be a very bad bandit method. What is the point here?
>
> Answer:
> We clarify that the B-batch bandit algorithm (Algorithm 3) uses data from all previous batches, not just the current batch. As the reviewer notes, discarding previous batch data would invalidate its nature as a bandit algorithm. While using all previous batch data makes the observations non-iid, our key contribution is that even with this non-iid data, the sample means remain asymptotically normal. We do so by verifying that all our bandit algorithms discussed in the paper satisfy the stability property due to Lai and Wei 1982 (Lemma 1).  We thank the reviewer for highlighting the need to better explain the algorithm's bandit nature and for noting the proof typos. We will address both in the final version.
>
> Comment: A strong paper would study the tradeoff between CLT friendliness and performance; nothing of this form is attempted here. The results, while they appear to be correct, do not require much effort.
>
> Answer:
> We appreciate this valuable suggestion about studying the tradeoff between CLT properties and performance. While our appendix presents numerical simulations demonstrating that the sample means closely follow normal distributions, we acknowledge that a deeper analysis of this tradeoff would strengthen the paper. In the final version, we will expand our numerical studies to include detailed coverage analysis of the confidence intervals. Following the many reviewer's suggestion we also intend to include the stability of K-arm bandit model when an analogous version of Algorithm 3 is used.
>
> Concretely, Theorem 3 in our paper has a natural K-arm generalization. We can show that for a K-arm bandit problem with B-batches, the elimination algorithm remains stable for K-arms, and we can quantify the limiting distribution of all K-arms. The extension to K-arms for the Batched Bandits can be implemented as follows:
> At each round b, we maintain a set of active arms A_b – the set of arms to be pulled in that round. A_b is defined as:
> A_b = {a ∈ A_{b-1}: UCB_of_arm_a > max_{a' ∈ A_{b-1}}[LCB of arm a']}
> In other words, we compare the Upper Confidence Bound of each arm with the maximum Lower Confidence Bound among arms in the previous active set A_{b-1}. An arm remains in the new active set A_b if its UCB exceeds this maximum LCB. We can prove this algorithm is stable, and consequently, all K arm-means are asymptotically normal. We will include this result in the final version of the paper. Please let us know if you would like us to provide additional details on the proof techniques or main results for the K-armed case in the review response.

---

> > ### Comment · Reviewer_qeHo · 2024-11-25
> >
> > I appreciate the authors' willingness to improve their paper, but in my opinion, it will be best for the community if the revision goes through the reviewing process again.

---

### Author Response · Authors · 2024-11-19
**Contribution and Extension to K-armed bandits.**

We thank the reviewers for their valuable feedback. Multiple reviewers raised two fundamental questions:

1. The contribution of our work relative to existing work
2. The potential extension of our results to K-armed bandits

We address these central questions below before responding to specific reviewer comments.

Comparison with existing work:
We want to clarify our paper's contribution in relation to existing work, particularly Khamaru and Zhang 2024 [KZ24]. The notion of stability originates neither from our work nor from KZ24. As noted in our paper (lines 132-135) and in KZ24's abstract, this concept is derived from Lai and Wei's seminal 1982 work, specifically Theorem 3. Lai and Wei established this stability notion for contextual bandits and proved that least square estimates become asymptotically normal when stability conditions are met. While Lai and Wei didn't explicitly use the term 'stability,' all essential conditions appear in their 1982 work. We included this stability notion, with a simple self-contained proof, because it remains understudied in bandit literature. This understudied nature stems from Lai and Wei not identifying any bandit algorithms satisfying such stability conditions, and it wasn't obvious whether any bandit algorithm would satisfy them.

KZ24's main contribution was proving that the popular UCB algorithm satisfies this Lai-Wei stability. However, UCB or epsilon-UCB becomes unstable when collecting bandit data in few batches. This is demonstrated in section 3 of paper [2] and more generally in Zhang et al. 2020 [3]; we highlight one such result in Lemma 2 of our paper.

Our paper's key contribution is demonstrating that elimination-based algorithms maintain stability when data is collected in few batches. Theorems 1 and 2 prove this for a 2-batch algorithm, while Theorem 3 extends this stability to the B-batch algorithm in the 2-armed case."

=======================================================

Extension to K-armed bandit:

Theorem 3 in our paper has a natural K-arm generalization. We can show that for a K-arm bandit problem with B-batches, the elimination algorithm remains stable for K-arms, and we can quantify the limiting distribution of all K-arms. The extension to K-arms for the Batched Bandits can be implemented as follows:

At each round b, we maintain a set of active arms A_b – the set of arms to be pulled in that round. A_b is defined as: A_b = {a ∈ A_{b-1}: UCB_of_arm_a > max_{a' ∈ A_{b-1}}[LCB of arm a']}

In other words, we compare the Upper Confidence Bound of each arm with the maximum Lower Confidence Bound among arms in the previous active set A_{b-1}. An arm remains in the new active set A_b if its UCB exceeds this maximum LCB. We can prove this algorithm is stable, and consequently, all K arm-means are asymptotically normal. We will include this result in the final version of the paper. Please let us know if you would like us to provide additional details on the proof techniques or main results for the K-armed case in the review response.


[2] Bibaut, Aurélien, and Nathan Kallus. "Demistifying Inference after Adaptive Experiments." arXiv preprint arXiv:2405.01281 (2024).
[3]Zhang, Kelly, Lucas Janson, and Susan Murphy. "Inference for batched bandits." Advances in neural information processing systems 33 (2020): 9818-9829.

---

### Meta-Review · Area_Chair_DHnf · 2024-12-20

**Metareview:**

Major concerns about the paper being ready for publication have been raised. These include concerns about the lack of polishing of the text and paper, and, more importantly, about the depth of the analysis. Reviewer qeHo raises here a number of concerns which are, in my opinion, a great starting point for a major revision of the paper.

**Additional Comments On Reviewer Discussion:**

Reviewers read the rebuttal and replied to author comments. Overall, the discussion was solid but could not alleviate the concerns of the reviewers.

---

### Decision · Program_Chairs · 2025-01-22

Reject